# Percutaneous pericardial catheter drainage for symptomatic uremic pericardial effusions with narrow safety margins

**Soon Bae Kim[1], Eun Hye Yang[1], Ji Hoon Shin[2]***

**1** Division of Nephrology, Department of Internal Medicine, University of Ulsan College of Medicine, Asan Medical Center, Seoul, Korea, **2** Department of Radiology, University of Ulsan College of Medicine, Asan Medical Center, Seoul, Korea

* jhshin@amc.seoul.kr

## Abstract

### Background

Percutaneous pericardial catheter drainage (PCD) for pericardial effusion is generally known to be limited by the high risk associated with effusions that are less than 10 mm thick. The objective was to report cases who underwent percutaneous PCD for symptomatic uremic pericardial effusions, which were less than 10 mm thick after cardiologist declined to perform the PCDs because of the narrow safety margins.

### Materials and methods

Thirteen consecutive cases (11 patients) (median age, 56 years, range, 31–83) with symptomatic uremic pericardial effusion (thickness <10 mm) affecting the pericardial space anterior to the right ventricle underwent ultrasound- and fluoroscopy-guided percutaneous PCD between September 2015 and April 2022. Information regarding the clinical criteria, echocardiographic features, PCD details, nature of effusion, and outcomes, including success and complications were retrospectively evaluated.

### Results

Pigtail catheter (8.5-Fr) insertion was successful for all patients, with a median procedure time of 7 minutes (range 4~12) without procedure-related complications. The median amount of drainage on the day of PCD was 700 mL (range, 250–1100). The median duration of catheter indwelling was 5 days (range, 1~32). In one case, the catheter was removed after 1 day due to chest pain. For all patients, pericardial effusion evacuation was achieved with relief of associated symptoms, representing 100% clinical success.

### Conclusion

Percutaneous PCD may be safely performed for patients with symptomatic uremic pericardial effusions and narrow safety margins of less than 10 mm.

**Data Availability Statement:** All relevant data are within the paper and its Supporting Information files.

**Funding:** he authors received no specific funding for this work.

**Competing interests:** The authors have declared that no competing interests exist.

## Introduction

Pericarditis is observed in advanced uremia. Uremic pericarditis is an absolute indication for the urgent initiation of dialysis or for intensification of the dialysis prescription in those already receiving dialysis. A pericardial drainage procedure should be considered in patients with recurrent pericardial effusion, especially with echocardiographic signs of impending tamponade [1].

Although intensive dialysis is the first line strategy in uremic patients with pericardial effusion [2], previously we lost two patients with symptomatic pericardial effusion without signs of tamponade with narrow safety margin in spite of intensive dialysis (Table 1). In Case 1 who were on continuous ambulatory peritoneal dialysis (CAPD) for 7 years, the patient died of pericardial tamponade in spite of continuing CAPD 4 times a day plus 5 times of hemodialysis during 13 days. Intensification of hemodialysis (5–7 days per week) is only effective approximately 50% of the time in dialysis-associated pericarditis (dialysis over 8 weeks) [3]. Although uremic pericarditis (prior to or within 8 weeks of initiation of dialysis) responds extremely well to initiation of dialysis [3], our Case 2 died of pericardial tamponade in spite of hemodialysis 7 times during 10 days. In addition, there is no guideline how long we should do this "intensive dialysis". The reason for mentioning these two patients is to show that untreated symptomatic pericardial effusion with a narrow safety margin can be fatal. Thereafter, we actively proceed to pericardiocentesis in patients with symptomatic uremic pericardial effusion without safety margin even in the absence of tamponade. Another reason for active pericardiocentesis is early differential diagnosis of other causes of pericardial effusion, such as malignancy, bacterial or tuberculosis.

Since the surgical subxiphoid approach for pericardial effusion drainage was first described in 1829, several additional methods for surgical drainage have been proposed [4–6]. The gold standard remains the surgical subxiphoid approach, which allows for the collection of fluid samples, pericardial biopsy, and pericardial drainage [7]. In 1986, Kopecky and colleagues reported the first percutaneous pericardiocentesis series with multiple subsequent reports characterizing the relative safety and efficacy of a percutaneous approach [4,8].

**Table 1. Patient characteristics and echocardiography of the two dead patients without PCD.**

| Case Number | 1 | 2 |
|---|---|---|
| Gender / Age | M/67 | M/71 |
| Duration of Dialysis | 7 years (PD) | 4 days (HD) |
| Grade of Dyspnea | NYHA III | NYHA III |
| Orthopnea | Yes | Yes |
| Other chest symptom | No | No |
| RV anterior (mm) | 9 | 4 |
| LV posterior (mm) | 37 | 8 |
| LV lateral (mm) | N/A | 6 |
| IVC plethora | No | No |
| Respiratory variation of mitral valve | N/A | N/A |
| Ejection fraction (%) | 58 | 58 |
| Management | Daily PD+HD 5 times/13 days | HD 7 times/10days |
| Echo before death | Massive pericardial effusion | Massive pericardial effusion |
| Time from initial Echo to Death | 13 days | 6 days |

PCD = pericardial catheter drainage; PD = peritoneal dialysis; HD = hemodialysis; NYHA = New York Heart Association; RV = right ventricle; LV = left ventricle; IVC = inferior vena cava; N/A = not available.

However, percutaneous pericardial catheter drainage (PCD) for pericardial effusion is generally known to be limited by the high risk associated with effusions that are less than 10 mm thick [2,7,9,10]. We report 13 cases (11 patients) who underwent ultrasound- and fluoroscopy-guided percutaneous PCD for symptomatic uremic pericardial effusions, which were less than 10 mm thick after cardiologist declined to perform the PCDs because of the narrow safety margins.

## Materials and methods

The study was approved by the institutional review board of Asan Medical Center (IRB No. 2020–1522), which waived the requirement for informed consent due to the retrospective nature of this study.

### Patients

Thirteen consecutive cases (11 patients) who were referred for percutaneous PCD between September 2015 and April 2022 were included in this study. The clinical characteristics of the 13 cases are summarized in Table 2. Nine cases were men, and the overall age range was 31 to 83 years (median, 56 years). Seven cases had been undergoing hemodialysis (median 2 years, range 1 day~7years) and 1 case on peritoneal dialysis (2.5 years). Modification of Diet in Renal Disease-estimated glomerular filtration rates (MDRD-eGFRs) in five non-dialysis patients ranged from 9 to 29 (median 19) mL/min/1.73m$^2$.

All 13 cases had dyspnea; one was classified as New York Heart Association (NYHA) class IV (dyspnea at rest), six were NYHA class III (less than ordinary physical activity caused symptoms), four were NYHA class II (ordinary physical activity caused symptoms) and two was NYHA class I (more than ordinary physical activity caused symptoms).

We provided the echocardiographic data on the absence/presence of tamponade/pre-tamponade diagnostic signs such as "respiratory variation of mitral inflow or diastolic collapse of right atrium or right ventricle in addition to inferior vena cava plethora in Table 2. Signs of pericardial tamponade were found in 9 cases; definitely in 7 and equivocally in 2. Eight cases had large effusion on the echocardiographic findings (circumferential effusion with the echo-free space greater than 20 mm at the widest point. All six patients who had equivocal or no evidence of tamponade had echocardiographically large pericardial effusion. We could not find the results of paradoxical pulse.

Chest CT was performed in 9 out of 13 cases to make sure encircling nature of pericardial effusion, because the presence of loculation further reduces the chance of success of percutaneous interventions.

The time from detection of pericardial effusion to PCD was median 2 days (range, same day~8 months). PCD was performed within 8 days in 10 out of 13 cases. Echocardiography was not performed after PCD because there was a large amount of drainage and there was a dramatic improvement in symptoms and chest radiographs.

### Percutaneous PCD technique

All PCD procedures were performed by one interventional radiologist with 20 years of experience. After administration of 1% lidocaine to the skin and the deeper tissue of the subxiphoid area, a 21-gauge Chiba needle (Cook, Bloomington, IN, USA) was advanced via a subxiphoid approach under ultrasound and fluoroscopic guidance (Fig 1A). The Chiba needle was advanced into the skin at a cephalad angle, and the coronary, pericardial, and internal mammary arteries were avoided. Once the fluid-filled pericardial space was entered with the Chiba

**Table 2. Patient characteristics, echocardiography, and details of PCD.**

| Case Number | 3 | 4 | 5 | 6 | 7 | 8[#] | 9 | 10[##] | 11 | 12 | 13 | 14 | 15 |
|---|---|---|---|---|---|---|---|---|---|---|---|---|---|
| Gender / Age | F / 57 | M / 37 | M / 56 | M / 31 | M / 59 | M / 41 | F / 56 | M / 60 | M / 49 | M / 65 | F / 83 | M / 75 | F / 34 |
| Duration of Dialysis | 7 years (HD) | 2 weeks (HD) | No | 1 week (HD) | No | 4 years (HD) | 2.5 years (PD) | No | 2 years (HD) | 1 day (HD) | No | No | 2.5 years (HD) |
| MDRD-eGFR (ml/min/1.73m$^2$) | N/A | N/A | 15 | N/A | 26 | N/A | N/A | 19 | N/A | N/A | 29 | 9 | N/A |
| Dyspnea (NYHA) | III | II | III | II | III | III | I | II | II | III | III | IV | I |
| Orthopnea | Yes | Yes | Yes | No | Yes | No | No | No | No | Yes | Yes | Yes | No |
| Other chest symptoms | Chest pain | Hemoptysis | Chest pain | No | No | Chest pain | No | No | Chest pain | No | No | No | No |
| Date of Initial Echo | 20150903 | 20161129 | 20190510 | 20200204 | 20200106 | 20201031 | 20201211 | 20200818 | 20210803 | 20211012 | 20220205 | 20220208 | 20220401 |
| RV anterior (mm) | 8 | 9 | 7 | 8 | 4 | 9 | 4 | 6 | 8 | 6 | 5 | 6 | 9 |
| RA posterior (mm) | 20 | 21 | 26 | 15 | 15 | 15 | 16 | 18 | 19 | 17 | 14 | 14 | 14 |
| LV posterior (mm) | 16 | 12 | 7 | 23 | 22 | 14 | 25 | 22 | 12 | 23 | 22 | 18 | 24 |
| LV lateral (mm) | 16 | 16 | 20 | 20 | 18 | 18 | 22 | 19 | 13 | 14 | 15 | 15 | 22 |
| IVC plethora | Yes | Yes | Equivocal | No | No | Yes | No | No | Yes | Equivocal | No | Yes | No |
| Other signs of cardiac tamponade | N/A | Resp. variation of hepatic vein flow | No diastolic collapse of RA or RV | No resp. variation of mitral inflow | N/A | N/A | No diastolic collapse of RA or RV | Equivocal resp. variation of mitral inflow | No resp. variation of mitral inflow | Resp. variation of mitral inflow | N/A | No diastolic collapse of RA or RV | Diastolic collapse of RA or RV |
| Ejection fraction (%) | 68 | 60 | 60 | 58 | 56 | 43 | 58 | 47 | 50 | 69 | 61 | 60 | 67 |
| CT | N/A | N/A | Yes | Yes | Yes | Yes | N/A | Yes | Yes | Yes | Yes | Yes | N/A |
| Date of PCD | 20150908 | 20161207 | 20191111 | 20200205 | 20200825 | 20201102 | 20201211 | 20201217 | 20210805 | 20211012 | 20220205 | 20220208 | 20220405 |
| Time from detection of PE to PCD | 5 days | 8 days | 6 months | 1 day | 8 months | 2 days | Same day | 4 months | 2 days | 2 days | Same day | Same day | 4 days |
| Procedure time (min) | 10 | 4 | 5 | 5 | 7 | 11 | 8 | 7 | 8 | 12 | 7 | 6 | 12 |
| Nature of drainage | Bloody | Mild turbid | Clear | Clear | Clear | Bloody | Clear | Clear | Clear | Mild turbid | Mild turbid | Bloody | Mild turbid |
| Pericardial WBC (/uL) | 740 | 580 | 120 | 372 | 128 | 2,198 | 179 | 384 | 152 | 1298 | 708 | 659 | 660 |
| Drainage amount (cc)[###] | 250 | 700 | 1,110 | 850 | 740 | 420 | 310 | 510[####] | 690 | 510 | 980 | 985 | 850 |

*(Continued)*

**Table 2.** (Continued)

| Dyspnea after PCD | No | No | No | No | No | No | No | No | No | No | No | No | No |
|---|---|---|---|---|---|---|---|---|---|---|---|---|---|

PCD = pericardial catheter drainage; PD = peritoneal dialysis; HD = hemodialysis; MDRD-eGFR = Modification of Diet in Renal Disease-estimated glomerular filtration rate; NYHA = New York Heart Association; RV = right ventricle; RA = right atrium; LV = left ventricle; IVC = inferior vena cava; N/A = not available; Resp. = respiratory; PE, pericardial effusion; WBC = white blood cell.

[#] the same patient with 4

[##] the same patient with 7

[###] on the day of PCD

[####] not fully drained, pericardiectomy at 1 month later.

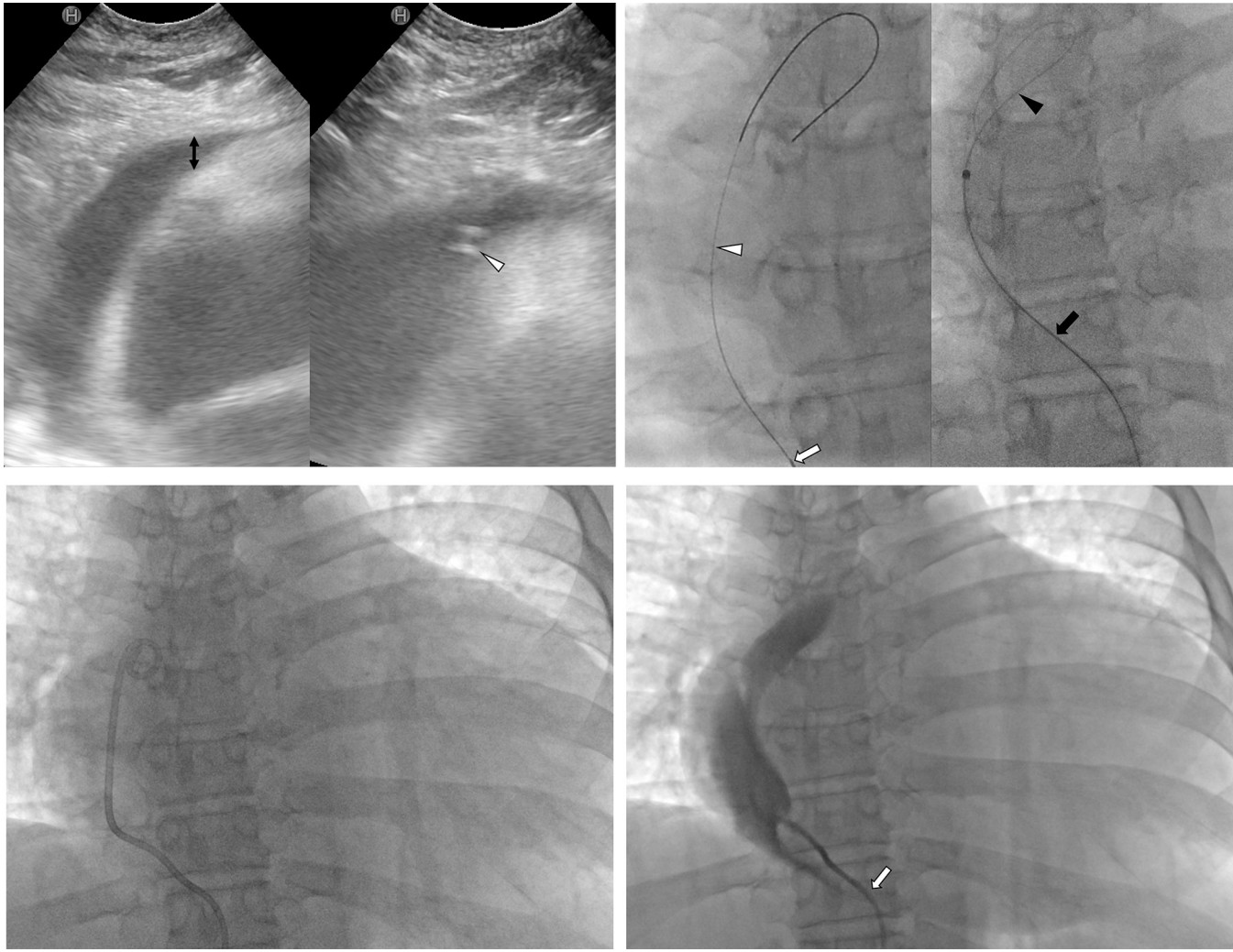

**Fig 1. A 59-year old male with dyspnea due to uremic pericardial effusion.** (A) The pericardial effusion thickness (black arrow) was 4 mm anterior to the right ventricle from the subxiphoid view of the ultrasound. A 21-gauge Chiba needle tip (arrowhead) is in the pericardial space on the ultrasound. (B) A 0.018-inch guidewire (white arrowhead) was introduced into the pericardial space through the Chiba needle (white arrow). The Chiba needle was exchanged for a 6-Fr Neff catheter (black arrow), through which a 0.035-inch stiff guidewire (black arrowhead) was advanced. (C) An 8.5-Fr pigtail catheter was inserted into the pericardial space. (D) The position of the catheter was confirmed to be in the pericardial space by injection of a contrast agent. The pericardial space into which the catheter enters is narrowed (arrow).

needle tip under ultrasound guidance, a 0.018-inch guidewire (Cook) was introduced into the pericardial space through the Chiba needle under fluoroscopic guidance (Fig 1B).

Then, the Chiba needle was exchanged for a 6-Fr Neff catheter (Cook). A 0.035-inch stiff guidewire (Terumo, Tokyo, Japan) was advanced through the Neff catheter (Fig 1B), and an 8.5-Fr pigtail catheter (Cook) was inserted into the pericardial space over the 0.035-inch guidewire (Fig 1C). After the effusion was manually drained, the position of the catheter was confirmed to be in the pericardial space by injection of a contrast agent (Fig 1D). The catheter was secured to the skin with 2–0 silk sutures and maintained in place until the amount of drainage was less than 30 mL/day [3].

## Data analysis and definitions

Medical records were reviewed to capture the clinical criteria, echocardiographic features, PCD details, nature of effusion, and outcomes, including success and complications.

Technical success was defined as the application of a draining catheter inside the pericardial space and drainage of the pericardial fluid. Procedure time was defined as the time from the start of lidocaine administration to the insertion of the drainage catheter. Clinical success of the technique was defined as the evacuation of the pericardial effusion and alleviation of symptoms associated with pericardial effusion.

Complications were divided into minor and major, according to the guidelines of the Society of Interventional Radiology. Minor complications were defined as requiring no additional treatment or hospitalization overnight for observation. Major complications were those that required therapy with minor hospitalization (<48 h), requiring major therapy, prolonged hospitalization (>48 h), or unplanned increase in the level of care, permanent adverse sequelae, or death [11].

## Results

Technical success was achieved, with insertion of all 8.5-Fr pigtail catheters in all 13 cases. For all 13 cases, the Chiba needle tip entered the target pericardial space accurately under ultrasound guidance (Fig 1A). The median procedure time was 7 minutes (range 4–12). There were no procedure-related complications.

The median amount of drainage on the day of PCD was 700 mL (range 250–1100) and 10 cases had large pericardial effusion (>500ml). In terms of appearance, six pericardial effusions were clear, four mildly turbid, and three were bloody. The median pericardial fluid white blood cell count was 580/μL (range, 120–2197). Other causes of pericardial effusion, such as malignancy, bacterial infection, and tuberculosis, were ruled out by cytology, bacterial culture, tuberculosis culture, and adenosine deaminase test.

The median duration of catheter indwelling was 5 days (1~32). In case 11, the catheter was removed after 1 day due to chest pain. For all patients, pericardial effusion evacuation was achieved with relief of associated symptoms, representing 100% clinical success, however, in case 10, the pericardial effusion recurred, and pericardiectomy was performed one month later.

## Discussion

To the best of our knowledge, this was the first study to attempt percutaneous PCD on patients with symptomatic pericardial effusions and narrow safety margins. Evacuation of the pericardial effusion was achieved with relief of dyspnea and other symptoms, achieving 100% clinical success. All 13 cases had safety margins less than 10 mm, and the narrowest safety margin was 4 mm in three patients.

Cardiologists have suggested that more observation or surgical treatment should be considered as PCD is technically difficult. However, there were practical problems such as a surgeon's tight schedule to consider surgical treatment, and there were cases where surgery could not be considered due to the rapid deterioration of the patient's situation. Therefore, it is considered as a didactic article that the indication of percutaneous PCD can be broadened.

If quantitation is desirable or required, a reasonable echocardiographic approach is to grade effusions as small, medium, or large, as determined by the size of the echo-free space surrounding the heart [9]. Large effusions (>500 mL) tend to be circumferentially visible; the echo-free space is greater than 20 mm at the widest point [9]. Our eight cases had echocardiographically large pericardial effusion and the amount of drained pericardial fluid was greater than 500cc in 10 cases. However, since the success of PCD depends on the width of the pericardial space to be accessed percutaneously, PCD is not always possible even with large effusions. In general, on cardiovascular magnetic resonance images, when the circumferential fluid-filled pericardial space width is less than 4 mm anteriorly to the right ventricle, the effusion is small, whereas a width greater than 5 mm indicates a larger effusion [12]. Therefore, there may be cases where the width of the pericardial space to be accessed percutaneously is less than 10 mm with large pericardial effusion, and percutaneous PCD is required.

Though size is relevant, the hemodynamic consequences of pericardial effusions are also related to the rate of fluid accumulation. For example, in Patient 3, only 250 mL (moderate effusion by amount but large effusion by echocardiography) caused NYHA grade III dyspnea and orthopnea. Rapidly increasing pericardial fluid quickly exceeds the limit of parietal pericardial stretch, causing a steep rise in pressure, which becomes even steeper as smaller increments in fluid cause a disproportionate increase in the pericardial pressure [12]. The bloody pericardial effusion was suspected to increase pericardial pressure rapidly in this patient.

In this study, when the pericardial space width was less than 10 mm at the right ventricular anterior surface, the subxiphoid approach was safe, and the procedure time was short (about 7 minutes). With the subxiphoid approach, the acceptable sonic window ensures that the puncture needle tip is accurately positioned within the pericardial space, enabling a fast and safe procedure. Many reports and guidelines describe an effusion thickness of at least 10 mm as an indication of percutaneous PCD [2,7,9,10]; therefore, this study's claim that PCD is possible even with effusions narrower than 10 mm requires further investigation with larger-scale studies. It is up to the attending physician whether to wait while doing intensive dialysis or to try pericardiocentesis in patients with symptomatic uremic pericardial effusions with narrow safety margins.

This study had several limitations. First, this study evaluated a small number of patients retrospectively. Second, percutaneous PCD was performed on uremic patients only. To definitively evaluate the effectiveness and safety of PCD for patients with symptomatic pericardial effusions and narrow safety margins, more research will be needed.

In conclusion, percutaneous PCD may be safely performed for patients with symptomatic uremic pericardial effusions and narrow safety margins of less than 10 mm.

## Author Contributions

**Conceptualization:** Soon Bae Kim, Eun Hye Yang, Ji Hoon Shin.

**Data curation:** Soon Bae Kim, Eun Hye Yang, Ji Hoon Shin.

**Formal analysis:** Soon Bae Kim, Eun Hye Yang, Ji Hoon Shin.

**Investigation:** Soon Bae Kim, Eun Hye Yang, Ji Hoon Shin.

**Methodology:** Soon Bae Kim, Eun Hye Yang, Ji Hoon Shin.

**Project administration:** Ji Hoon Shin.

**Resources:** Ji Hoon Shin.

**Supervision:** Ji Hoon Shin.

**Validation:** Soon Bae Kim, Eun Hye Yang, Ji Hoon Shin.

**Visualization:** Soon Bae Kim, Ji Hoon Shin.

**Writing – original draft:** Soon Bae Kim, Eun Hye Yang, Ji Hoon Shin.

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
