## [Decision Letter · Decision Letter 0]

28 Apr 2022

PONE-D-21-26425

Percutaneous Pericardial Catheter Drainage for Symptomatic Uremic Pericardial Effusions with Narrow Safety Margins

PLOS ONE

Dear Dr. Shin,

Thank you for submitting your manuscript to PLOS ONE. After careful consideration, we feel that it has merit but does not fully meet PLOS ONE’s publication criteria as it currently stands. Therefore, we invite you to submit a revised version of the manuscript that addresses the points raised during the review process.

We look forward to receiving your revised manuscript.

Kind regards,

Arturo Cesaro, MD

Academic Editor

PLOS ONE

Journal Requirements:

Additional Editor Comments (if provided):

You are invited to consider the reviewers' comments, reported at the end of this letter, and to revise your manuscript accordingly. In the letter accompanying your resubmission, please explain your response to each of the comments. Please observe the word count and citation style. For further details, please consult the Instructions for Authors on the website

Reviewers' comments:

Reviewer's Responses to Questions

**Comments to the Author**

1. Is the manuscript technically sound, and do the data support the conclusions?

Reviewer #1: Partly

Reviewer #2: Partly

2. Has the statistical analysis been performed appropriately and rigorously? 

Reviewer #1: N/A

Reviewer #2: N/A

3. Have the authors made all data underlying the findings in their manuscript fully available?

Reviewer #1: Yes

Reviewer #2: Yes

4. Is the manuscript presented in an intelligible fashion and written in standard English?

Reviewer #1: Yes

Reviewer #2: No

5. Review Comments to the Author

Reviewer #1: Authors describe a case series of patients with symptomatic large uremic pericardial effusion with narrow safety margins for pericardiocentesis with subxyphoid access. Patients underwent pericardiocentesis under echocardiographic and fluoroscopic guidance. The manuscript is written in clear standard English. It is a retrospective decription of the cases.

There are some major revisions required: to provide echocardiographic data on the absence/presence of tamponade/pre-tamponade diagnostic signs; to better explain indications to procede to pericardiocentesis even in the absence of tamponade or of the demostration of resistance to treatment (why intensive dialysis wasn't the prefferred first line strategy in all the cases? - Am J Med Sci. 2003 Apr;325(4):228-36); to explain why it was not considered to use approach other than subxyphoid, since with echocardiographic guidance it is possibile to perform safely apical and parasternal approaches to reach the maximum amount of fluid to overcome the absence of standard safety margins in the subxyphoid approach; to remove references to UpToDate website and specifically cite papers that support the statements.

Reviewer #2: Dear Authors,

I read your article with great interest. Pericardial effusion can occur in many metabolic and oncological diseases. It is a condition with a high risk of recurrence if not treated with an appropriate approach. We have interventional and surgical options for intervention. There are many studies comparing these two methods in terms of safety and effectiveness. I would like to explain my criticisms about the study you have done, item by item, in the continuation of the article:

1) In the study, you have presented the results of seven patients. It means that two patients had to be re-intervented with PCD. In addition to that, the 8th patient had undergone surgery one month after PCD. Actually, these outcomes indicate that PCD was a permanent solution for only four patients. In your article, you mentioned that the cardiologists had rejected the PCD because of the narrow safety margin. Cardiologists and Cardiothoracic Surgeons are the professionals who manage the process of heart-related medical conditions. Cardiologists might reject the PCD because of the high risk but they ought to suggest a better and less risky way to fix the problem. In your study, you did not mention what their suggestions were instead of PCD.

Every high-risk intervention could be performed successfully by some talented medical associates. Having abilities to perform high-risk interventions is not a scientific justification to do it. Patients' safety and permanency of the therapy must be the main goals of every healthcare professional.

2) In the guideline, the recommendation for pericardial effusion with a known etiology is primarily to treat the underlying disease. Pericardial effusion is expected to decrease in this way. It was stated in your study that 3 patients did not receive any renal replacement therapy prior to PCD. One of these patients had also anasarca. Why didn’t you perform dialysis for these three patients primarily?

3) The guideline recommends intervention for chronic (>3 months) and large pericardial effusions. There are only two patients who showed signs of pericardial tamponade(IVC plethora, 2 and 9). Dyspnea, hemoptysis, and orthopnea occur in pulmonary edema due to volume overload. These are not specific findings of pericardial tamponade.

How long was the pericardial effusion followed in these patients? How long was the medical treatment given? Was the intervention planned as soon as pericardial effusion was detected? Were there other findings for pericardial tamponade, such as an inspiratory change in blood pressure?

4) In echocardiographic measurements, the distribution of pericardial effusion is highly variable in a few cases. Was further examination(CT or CMR) performed considering the possibility of loculations? Because the presence of loculation further reduces the chance of success of percutaneous interventions.

5) Finally, I think that a 9-case study is not sufficient to recommend a high-risk intervention as "safely feasible". If you could increase the number of cases and datas, it might be a considerable study.

6. PLOS authors have the option to publish the peer review history of their article (what does this mean?). If published, this will include your full peer review and any attached files.

Reviewer #1: No

Reviewer #2: No

---

## [Author Response · Author response to Decision Letter 0]

11 May 2022

Response was submitted as an attachment file.

---

## [Decision Letter · Decision Letter 1]

31 Aug 2022

PONE-D-21-26425R1Percutaneous Pericardial Catheter Drainage for Symptomatic Uremic Pericardial Effusions with Narrow Safety MarginsPLOS ONE

Dear Dr. Shin,

Thank you for submitting your manuscript to PLOS ONE. After careful consideration, we feel that it has merit but does not fully meet PLOS ONE’s publication criteria as it currently stands. Therefore, we invite you to submit a revised version of the manuscript that addresses the points raised during the review process.

We look forward to receiving your revised manuscript.

Kind regards,

Wisit Kaewput, MD

Academic Editor

PLOS ONE

Journal Requirements:

Additional Editor Comments:

I really appreciate your hard work and tremendous improvement on methods and results. All comments had appropriate addressed. However some minor concern should be address as following. I invite you therefore to respond to the reviewer(s)' comments and revise your manuscript accordingly. I hope you will take the opportunity to revise your manuscript

Reviewers' comments:

Reviewer's Responses to Questions

**Comments to the Author**

1. If the authors have adequately addressed your comments raised in a previous round of review and you feel that this manuscript is now acceptable for publication, you may indicate that here to bypass the “Comments to the Author” section, enter your conflict of interest statement in the “Confidential to Editor” section, and submit your "Accept" recommendation.

Reviewer #2: (No Response)

Reviewer #3: All comments have been addressed

2. Is the manuscript technically sound, and do the data support the conclusions?

Reviewer #2: Partly

Reviewer #3: Yes

3. Has the statistical analysis been performed appropriately and rigorously? 

Reviewer #2: Yes

Reviewer #3: Yes

4. Have the authors made all data underlying the findings in their manuscript fully available?

Reviewer #2: Yes

Reviewer #3: Yes

5. Is the manuscript presented in an intelligible fashion and written in standard English?

Reviewer #2: Yes

Reviewer #3: Yes

6. Review Comments to the Author

Reviewer #2: Dear Authors,

Thank you for your response. In your study, there are still some sections that must be revised.

1- In your revised paper, I couldn't find the answer to "what was the cardiologists' suggestion instead of the PCD?". In my opinion, surgery(pericardial window) would have been a safer and more permanent treatment option for all the patients in your study. I think that the cardiologists suggested the surgery instead of PCD.

2- Patient 1 had a large pericardial effusion upon admission (Posterior of LV, 37mm). Surgery would be the option for treatment. The PCD is not a feasible method for this case.

Patient 2 didn't have pericardial effusion that needed an aggressive intervention.

In pre-mortem echo findings, you have mentioned massive pericardial effusion. What are your measurements?

These two patients are not similar cases. They can't be a medical basis for your study.

3- Could you please mention the post-interventional echo findings in Table 2?

Reviewer #3: Authors have addressed the reviewers' comments a properly. I have one minor change to suggest - the date format in table 1 and table 2 (yearmonthdate) are not in universal format. In fact, presenting the exact date in the manuscript might not be appropriate because this is patient identifiable information.

7. PLOS authors have the option to publish the peer review history of their article (what does this mean?). If published, this will include your full peer review and any attached files.

Reviewer #2: No

Reviewer #3: No

---

## [Author Response · Author response to Decision Letter 1]

25 Sep 2022

Dear Dr Wisit Kaewput

 We have received your decision letter regarding our manuscript PONE-D-21-26425R1, entitled "Percutaneous Pericardial Catheter Drainage for Symptomatic Uremic Pericardial Effusions with Narrow Safety Margins". We were happy to learn that our manuscript will be reconsidered for publication pending the completion of revisions. 

 We have changed the manuscript as much as possible according to the reviewer’s comments. We wish to thank you and the reviewers for your valuable comments and helpful suggestions which contributed significantly to the revision of our manuscript. We ensured that our manuscript met PLOS ONE’s style requirements and Full ethics statement was included in the ‘Methods’ section. We are also sending our responses to the comments of the reviewers in a point-by-point fashion. 

We ensure that our manuscript met PLOS ONE's style requirements, including those for file naming. We also ensure that our manuscript included our full ethics statement in the ‘Methods’ section of our manuscript file.

Response to Second reviewer

REV 2-1: In your revised paper, I couldn't find the answer to "what was the cardiologists' suggestion instead of the PCD?". In my opinion, surgery(pericardial window) would have been a safer and more permanent treatment option for all the patients in your study. I think that the cardiologists suggested the surgery instead of PCD.

☞ We should have written the cardiologists' suggestion. We added the following into Discussion section (Page 8, Lines 160~164). 

“Cardiologists have suggested that more observation or surgical treatment should be considered as PCD is technically difficult. However, there were practical problems such as a surgeon's tight schedule to consider surgical treatment, and there were cases where surgery could not be considered due to the rapid deterioration of the patient's situation. Therefore, it is considered as a didactic article that the indication of percutaneous pericardial catheter drainage can be broadened.”

REV 2-2: Patient 1 had a large pericardial effusion upon admission (Posterior of LV, 37mm). Surgery would be the option for treatment. The PCD is not a feasible method for this case. 

Patient 2 didn't have pericardial effusion that needed an aggressive intervention.

In pre-mortem echo findings, you have mentioned massive pericardial effusion. What are your measurements?

These two patients are not similar cases. They can't be a medical basis for your study. 

☞ There is an interval of 13 or 6 days between initial echocardiography and death. Bedside echocardiography just before death did not show any detailed measurements, but there was a lot of pericardial effusion. In case of patient 2, the amount of pericardial effusion was not large on the initial echocardiography, but clinical symptoms such as dyspnea and orthostatic respiration required treatment.

We have added the following sentence in Introduction section (Page 3, line 54, 55) for clarity. 

“The reason for mentioning these two patients is to show that untreated symptomatic pericardial effusion with a narrow safety margin can be fatal.” 

REV 2-3: Could you please mention the post-interventional echo findings in Table 2?

☞ We did not perform post-interventional echo in most cases, because there were large amounts of drainage and dramatic improvements in symptoms and chest radiographs. We added the following in the Materials and Methods section (Page 6, Lines 103~104).

“Echocardiography was not performed after PCD because there was a large amount of drainage and there was a dramatic improvement in symptoms and chest radiographs.”.

Response to Third reviewer

REV 3-1: Authors have addressed the reviewers' comments a properly. I have one minor change to suggest - the date format in table 1 and table 2 (yearmonthdate) are not in universal format. In fact, presenting the exact date in the manuscript might not be appropriate because this is patient identifiable information.

☞ We agree with you. ‘Time from initial echo to death’ information is sufficient. We have deleted the date information for ‘Date of initial echo’ and ‘Date of death’. 

Thank you very much

---

## [Editor Report · Decision Letter 2]

10 Oct 2022

Percutaneous Pericardial Catheter Drainage for Symptomatic Uremic Pericardial Effusions with Narrow Safety Margins

PONE-D-21-26425R2

Dear Dr. Shin,

We’re pleased to inform you that your manuscript has been judged scientifically suitable for publication and will be formally accepted for publication once it meets all outstanding technical requirements.

Kind regards,

Wisit Kaewput, MD

Academic Editor

PLOS ONE

Additional Editor Comments (optional):

The authors addressed all previous comments. I have no additional concern.
---

## [Editor Report · Acceptance letter]

20 Oct 2022

PONE-D-21-26425R2 

Percutaneous Pericardial Catheter Drainage for Symptomatic Uremic Pericardial Effusions with Narrow Safety Margins 

Dear Dr. Shin:

I'm pleased to inform you that your manuscript has been deemed suitable for publication in PLOS ONE. Congratulations! Your manuscript is now with our production department. 

Kind regards, 

on behalf of

Dr. Wisit Kaewput 

Academic Editor

PLOS ONE